# Hazards of Activation of Cryogenic Processes in the Arctic Community: A Geopenetrating Radar Study in Lorino, Chukotka, Russia

**Oleg Tregubov [1], Gleb Kraev [2,3,*] and Aleksey Maslakov [4]**

[1]  North-East Interdisciplinary Scientific Research Institute, Far-East Branch, Russian Academy of Sciences, Anadyr 686710, Russia; tregubov2@yandex.ru

[2]  Institute of Physicochemical and Biological Problems in Soil Science of the Russian Academy of Sciences, Pushchino 142290, Russia

[3]  Faculty of Science Earth and Climate, Vrije Universiteit Amsterdam, 1081 HV Amsterdam, The Netherlands

[4]  Faculty of Geography, Lomonosov Moscow State University, Moscow 119991, Russia; alekseymaslakov@yandex.ru

*  Correspondence: kraevg@gmail.com

**Abstract:** The subsurface structure of permafrost is of high significance to forecast landscape dynamics and the engineering stability of infrastructure under human impacts and climate warming, which is a modern challenge in Arctic communities. Application of the non-destructive method of a geo-penetrating radar (GPR) survey is a promising way to study it. In this paper, we provide the study program, which could be used for the planning and monitoring of measures of adaptation of Arctic communities to environmental changes. Etalons of correlated radargrams and archive geological data were compiled to interpret changes in the permafrost composition from a grid of 5–10 m GPR transects in Lorino. Here, we show the application of GPR to reconstruct and predict hazards of activation of cryogenic processes from the spatial variability in the sequence of layers of permafrost. The indicator layers and variables for cryogenic processes were as follows: the thawed layer bottom, thicknesses and depths of the ice-rich transient layer at the bottom of the active layer, massive ice bodies, and frost-susceptible sediments. The permafrost degradation in Lorino has declined due to improvements in the maintenance of infrastructure and local permafrost aggradation in relic taliks after ground filling applications. However, the hazards of heaving and thermokarst remain for the built-up area.

**Keywords:** geohazards; georadar; near-surface; frost heaving; thermokarst; permafrost degradation; environmental change; GPR survey; geological and geophysical etalons; active layer; Arctic community

## 1. Introduction

Reports on the growing destruction and deformation of infrastructure has recently emerged from different inhabited locations in the Arctic [1–7]. Locally, up to 80% of buildings in large industrial centers have become deformed [4]. Recurrent deformations of infrastructure in spite of the application of different geotechnical solutions are common [8]. The reasons for this not only include the extreme conditions to which the construction materials are exposed and high costs of the maintenance. They are also caused by alterations of permafrost, which often acts as a basement for foundations. Its high vulnerability to increasing air temperature in the Arctic, which is predicted to continue over the 21st century [9], is widely recognized to be a reason for negative consequences to society [10,11], currently or in the future, and represents a natural hazard.

Despite the fact that the practice of regulated construction in the permafrost zone dates back to the mid-1950s, changes in heat and moisture conditions due to climate tendencies are not accounted for in building standards [12,13]. Another issue relates to uncertainties of subsurface conditions, which are higher in the permafrost zone than outside it because of the high variability in patterns and volumes of ground ice. Ground ice is not only represented by massive ice (conventional term [14]), but also by relatively thinner cryostructures. Together, they form a highly variable and thus hardly predictable ice content along an engineering object, causing unequal responses of permafrost to changes in local heat and moisture exchange during construction and exploitation. The growth of ground ice inclusions results in the frost heaving of soils and erected infrastructure. The thawing results in subsidence, and in the worst cases, in thermokarst. An unequal response of permafrost leads to deformations and destructions, which cause evolving disastrous consequences, affecting social security [4].

Unlike any other cryospheric object, changes to permafrost cannot be easily observed [9]. The application of a geo-penetrating radar (GPR) to permafrost studies has allowed researchers to reliably recognize thawed and frozen layers, massive ice, and rocks in continuous and discontinuous permafrost [15–20]. The stratigraphy of permafrost sections, comprising layers of different iciness [21,22] and salinity [23], was studied by combining a GPR survey with drilling. Several case studies have mapped aureoles of thawing in degrading permafrost and findings of massive ice and other permafrost features under engineering structures [24,25]. The effectivity of the method is based on the contrasting electrophysical properties of thawed and frozen sediments, ice, and aquifers. Limitations of the method include fading of the signal in waterlogged peaty soils of tundra or thick covering loams and saline sediments. The need to calibrate the instrument at every study site to unique geological sections is another constraint in the wide application of GPR.

The development of methods of subsurface sounding is a priority task under the predicted shrinkage of permafrost area, increasing permafrost temperature and active layer thickness [9]. They do not only non-destructively complement the spatial structure of permafrost deduced from "snapshots" of a sampled active layer thickness [19], exposed parts of geological sections, and survey drilling, which commute disturbance to the thermal regime of soils [26,27]; they are also the most effective way to monitor permafrost evolution under human impacts and environmental changes [28–31]. The combination of a GPR survey with available geological data is a powerful instrument of estimation of the dynamics of cryogenic processes, and the evolution of structure of permafrost [16,32].

With this case study, we share our experience of the application of GPR in a permafrost zone to reconstruct and predict cryogenic processes affecting the life of an Arctic community. Our goal was to assess the vulnerability of permafrost. To do so, we put forward the following objectives:

- To collect the available geological data from point sources (boreholes and exposure);
- To track the interfaces of a layer of thawed sediments and permafrost layers having contrasting electrophysical properties using GPR;
- To reconstruct recent geological processes, including hazardous thermokarst, thermal erosion, and frost heaving.

## 2. Materials and Methods

### 2.1. Study Site

The village of Lorino (65.4978° N, 171.7247° W) is located on the coast of the Mechigmen bay, Bering sea, on a fragment of a 24 m high marine terrace (Figure 1) in the mouth of Loren River. It is part of a narrow coastal plain surrounded by a mountain chain with heights of above 600 m a.s.l.

Lorino is characterized by a subarctic maritime climate [33], short and cool summer (mean temperature of July is +8 . . . +10 °C), and long and moderately cold winter (mean temperature of January is −16 . . . −20 °C). The mean annual air temperature had been growing steadily by 1.8 °C from 1970 to 2005, and then stabilized [34]. Mean precipitation of 220–240 mm was stable over the 1970–2019 time frame [35].

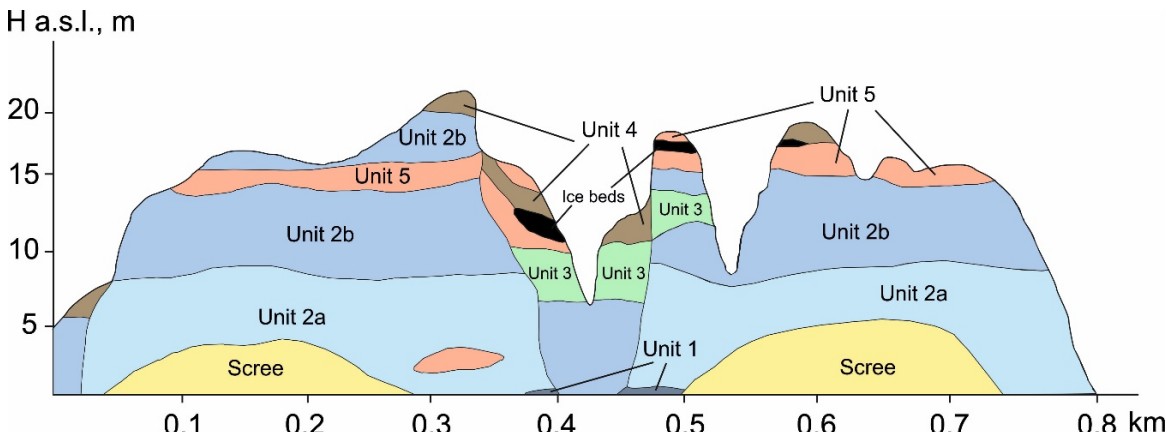

**Figure 1.** Diversity of the sediments comprising the geological section of the Lorino coastal bluff [36,37]. A description of sedimentary units is given in the text.

The geological section of the terrace is disclosed in a well-described exposure of the coastal bluff. It consists of deposits widely distributed on the Bering sea coasts [36]. A detailed description of the sediments can be found in our open access study [37]. Dark grey clayey loams and clays in the base of the bluff (Unit 1) are covered with sands (Unit 2a) banded with pebble and gravel-enriched layers (Unit 2b). The inlet of ice- and organic-rich saline fine sands (Unit 3) is disclosed in the center of the terrace, intersected by a deep gully. They are covered with grey saline sandy loams with gravel inclusions (Unit 5, 2–4 m thick), and peat dissected with ice wedges (Unit 4, up to 4 m thick). The territory of the settlement is regularly leveled with ground filling mainly composed of waste of coal burned on local steam shops.

Permafrost is continuous, mostly with a low ice content, and with the sparse occurrence of massive ice bodies [38]. Among the hazardous cryogenic processes and phenomena observed in Lorino are thermokarst, heaving, thermal erosion, and thermal abrasion, causing deformations of buildings and the destruction of infrastructure [37,39]. Furthermore, solifluction, kurums, and icings are common in the surroundings. The permafrost temperature measured in the boreholes by Chukotka Engineering Survey Enterprise (ChESE) was −3.7 . . . −4.5 °C in the 1990s. Mean annual temperatures of soil by depths in 1992 were 0 °C at 1.5 m, −3 °C at 4.5 m. The Circumpolar Active Layer Monitoring plot in pristine tundra 2 km away from Lorino has been showing a gradual increase of the active layer thickness from 0.5 to 0.7 m over the last decade [40,41]. The ice-rich transient layer just below the active layer [42] was observed in the exposure. Subsidence and thermal erosion have caused the deformation of several buildings and part of the infrastructure.

The population of Lorino was 998 in 2018 [43]. The settlement occupies 9.43 $km^2$ of land, of which 0.27 $km^2$ is the built-up area, with 37 residential low-rise (up to four floors) buildings of the 1960s–2000s, with the modern ones on pile supports; two steam shops; a diesel power station; a first-aid post; several stores; two schools; and a playground [44]. Roads are unpaved. The population mainly consists of indigenous Chukchi people involved in traditional marine mammal hunting, fishing, reindeer herding, fur farming, community services, and trade.

## 2.2. Methods and Instruments

Widely available GPRs with antennas ranging from 10 to 1000 MHz exhibit little difference in terms of their technical structure, principles of operation, and capabilities. GPR uses broadband pulses of electromagnetic radiation in the wavelength range from 1 to 0.1 m and detects reflected signals from subsurface interfaces of layers differing by electrophysical properties. Such interfaces may correspond to differences in the composition of sediment horizons, boundaries of thawed sediments and permafrost, massive ice bodies, differences in saturation, or mineralization.

We used an OKO-2 (LogiS, Russia) GPR with a 400 MHz shielded antenna block because 50–400 MHz is the optimal frequency range for surveying permafrost laid by 0.5–2.5 m thick layers down to a 10 m depth [16,17,21,23,25,26,28,45–47]. A GPR survey was performed by the steady motion of an interconnected generator of electromagnetic waves and a detector, fixed at some distance from one another. The motion along the predefined transects was supported by sledges.

The program of the survey consisted of four steps detailed in the Section 2.2.1, Section 2.2.2, Section 2.2.3, Section 2.2.4, followed by the geocryological and geoenvironmental interpretation described in Section 3.3.

### 2.2.1. Reconnaissance Survey

We started with a visual description of the study area, using community design documents and Google Earth images, paying attention to building types and non-conformities of the surface. The test GPR survey was performed along the streets to adjust the intensity of radiation and mode of detection to a depth of penetration of 5–10 m, resolution of 100–200 ns, and acceptable noise level. Then, the etalon transects and 50–150 m long transects were set up with an interval of 50–100 m, referenced to the streets and sidewalks around built-up areas (Figure 2).

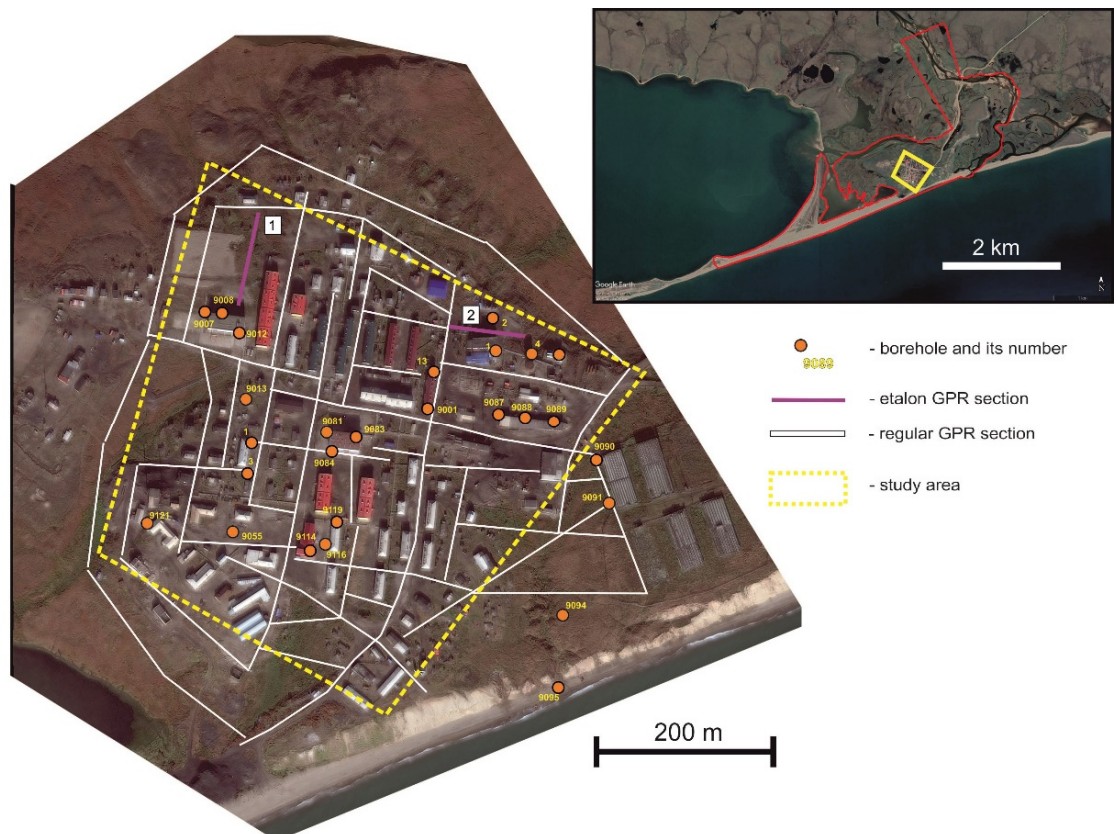

**Figure 2.** Grid of geo-penetrating radar (GPR) transects and survey boreholes in Lorino.

### 2.2.2. Etalon Geological and Geophysical Sections

Geological data were available from 10–15 m deep boreholes, which had been drilled in the 1990s by ChESE for projects of a new school and the diesel electric station. Mechanical, physical, and chemical features of sediments along the cores and the geological sections had been documented. Boreholes were selected for the etalons in order to cover the maximum diversity of geological sections typical for Lorino. Previous geophysical data for the study area were not found. GPR transects were collected along the mouths of the boreholes, corrected for the ground filling, which had buried the borehole mouths 0.5–1 m below the surface. Etalons were made by the correlation of geological sections and

geophysical sections (radargrams) and, thus, the electrophysical properties of the boundaries in the geological section were found. The most important one was the dielectric permittivity of medium ($\varepsilon$), which defines the velocity of wave propagation ($V$), the damping ratio, and delay of the reflected signal.

### 2.2.3. Collection and Processing of Geo-Penetrating Radar (GPR) Transects

Radargrams of GPR transects were collected in forward and return strokes with different resolutions to increase the overall depth of penetration to 10 m and allow for a more detailed survey of the top 5 m. Visually deformed sites were examined in more detail. A total of 36 GPR transects 50–150 m long were collected during 3 days in early winter, when the active layer had just started to freeze under mean daily temperatures of +5 . . . −1 °C (Figure 2).

The techniques of processing of radargrams are mostly analogous to those of seismograms because the laws of wave propagation and reflection are similar for electromagnetic and acoustic waves [23]. Radargrams were processed and graphically analyzed with GeoScan31 (Geotech, Russia) in order to amplify useful signals and attenuate the extraneous noise. Interpretation was based on the recognition of continuous reflection events corresponding to interfaces of media with different electric permittivity. Condensation or termination of the reflections gave images of the form of layers and lens-like objects in the section. Based on the $\varepsilon$ and $V$ estimated for each interface on the etalons, the boundaries on radargrams were interpreted as geological boundaries, and the distances between them as thicknesses of layers.

### 2.2.4. Spatial Modeling Based on the Grid of Transects

Thicknesses of layers and depths of interfaces were sampled from GPR transects at every 5 m, producing the grid of $565 \times 7$ data points superimposed on the village plan. Maps of depths (isochores) and thicknesses (isopachs) of the recognized layers were generated based on the grid, using Surfer 11.0 (Golden Software, Golden, CO, USA) to study spatial variabilities of the near surface structure of permafrost.

## 3. Results and Discussion

### 3.1. Etalons

The following interfaces were reliably recognized on radargrams of etalons: active layer bottom, bottom of ice-rich ground filling, massive ice beds, boundaries of buried organic-rich deposits 1 m thick at depths of 3–4 m, and the table of loams (Figure 3a,b). The layer of thawed sediments was recognized owing to the microplicated structure formed by multiple thin layers and folds originating from recurrent freezing-thawing cycles. Another distinctive feature was the interface of the ice-rich transient layer on the bottom. The ground-filling bottom was less contrasting, but still recognizable. Characteristic features of massive ice beds were a lens-like form, reaching a diameter of 30–50 m, and no secondary (parallel) reflections. Boundaries of organic-rich sediments were the most contrasting because of the large difference in dielectric permittivity compared to covering and embedding layers. Recognizing the table of loams at 6–8 m was the most difficult task.

Based on the sequence of layers, two 10 m-deep etalon sections were collected in eastern and western parts of the built-up area of the settlement (Figure 2).

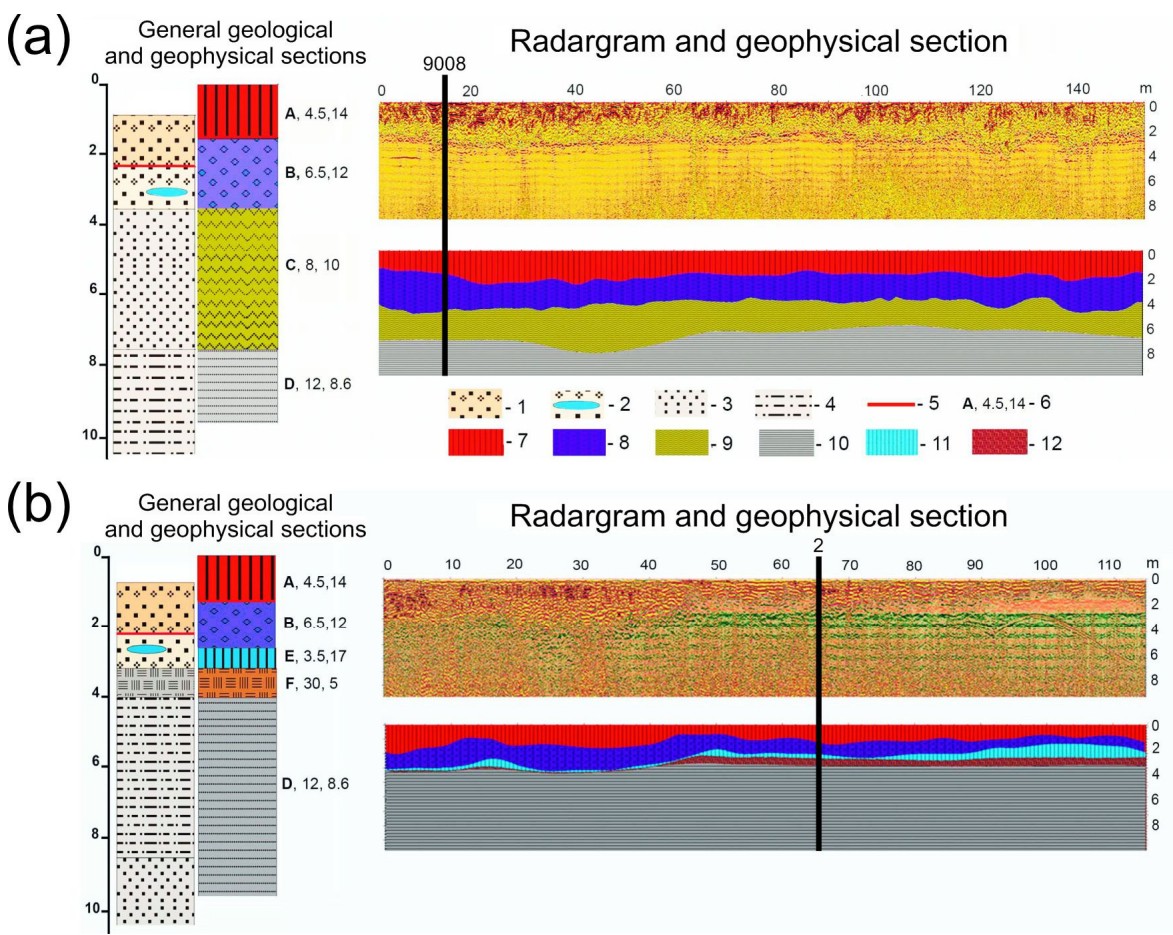

**Figure 3.** Etalons no. 1 (**a**) and no. 2 (**b**): correlation of geological sections disclosed by boreholes (9008 and 2) with radargrams. Key: 1—wet thawed coarse sand; 2—ice-rich gravelly sand with ice lenses; 3—cryotic fine sand; 4—ice-rich silty sand with bands of sandy loams; 5—bottom of the thawed layer; 6—layer index, dielectric permittivity ($\epsilon$), and wave velocity (V, cm ns$^{-1}$); 7—layer A; 8—layer B; 9—layer C; 10—layer D, 11—Layer E; 12—Layer F.

Etalon no. 1 was mainly composed of the sandy sediments of Units 2 and 3 (Figure 3a). Observed reflection interfaces were attributed to the frozen/thawed sediment interface, and the boundaries of alternating layers of different ice contents and sediment compositions. Four layers recognized on the radargram corresponded to the following geological layers:

- A—thawed layer (0.6–1.8 m thick), within the active layer and ground filling;
- B—ice-rich permafrost (total moisture content up to 35 wt%) with up to 1.4 m thick lenses of ice along the bottom (0.1–2.5 m), frozen ice-rich ground filling, and the transient layer, which were hard to recognize;
- C—cryotic sandy layer (total moisture content as low as 1 wt%, Unit 2b) (1–3 m), with blurred bottom;
- D—frozen loams and sandy loams (Unit 5).

Etalon no. 2 was composed of

- A—Layer A (1.0–1.2 m thick);
- B—Layer B (1.0–2.5 m);
- E—massive ice of various origins (up to 1.5 m);
- F—organic-rich frozen silts (0.5–1.3 m) with moisture content up to 65–70 wt% (Unit 4);

- D—Layer D, which included both the sandy loams and fine sands disclosed by the borehole at 8 m, but not recognized in the radargram.

The most prominent features of etalons were the bottom of the thawed layer (A), bottom of ice-rich deposits at the permafrost table (B), massive ice beds (E), and organic-rich silty deposits (F).

### 3.2. GPR Performance in Built-Up Areas of the Permafrost Zone

The study took place among buildings and infrastructure of the village. Wasted metal and construction materials were not rare in the section. Non-stationery high-frequency noise occurred from passing cars, work of the power station, etc. A GPR survey has at least several advantages in such conditions compared to microseismic sounding and different types of electric survey: compact and mobile instruments, controlled sensitivity of the instrument to technogenic noise, and short-term survey production.

Our application did not allow the features of permafrost, such as the particle composition, mechanical properties, salinity, moisture, ice, or organic matter content, to be measured for each layer. These features had been studied by ChESE using cores from several points (boreholes) in the village. Instead, we found spatial variations of the section and thicknesses of the layers in the whole built-up area.

Based on our decade-long experience of GPR studies in different communities of Chukotka [25,48], the following typical ranges of $\varepsilon$ could be reported for the sediments:

- 4–8 for the active layer;
- 6–9 for ice-rich fine sediments;
- 3.5–4.5 for ground ice;
- 20–40 for buried peat and organic-rich soils;
- 12–16 for frozen loams.

These values might be used as a reference in similar conditions when there is no other data for calibration. In the following section, we demonstrate how this data might be used for the reconstruction of geological processes and predicting the geohazards.

### 3.3. Permafrost Structure: Reconstruction and Predicting of Cryogenic Processes

The typical sections of permafrost, as interpreted from radargrams, are shown in Figure 4. Spatial patterns of the thickness and deformations of layers allowed the dynamics of cryogenic processes to be interpreted and the lands of Lorino, susceptible to different geohazards to be classified.

Sites with a uniform bottom of Layer A at 0.5–1 m had background values of seasonal thawing (Figure 4a), slightly deeper than that in pristine tundra nearby, likely due to different soil composition and drainage [41]. The thickening of Layer A up to 2–3 m associated with the thinning of Layer B marked taliks, indicating the active thermokarst (Figure 4a,b). The growth of ice-rich Layer B to 1–1.5 m thick was associated with the aggradation of permafrost by the freezing of ground filling (Figure 4b,c). The locally increased thickness of Layer F indicated heaving, observed at sites where the active layer reached the table of Layer F (Figure 4b). At sites where Layer B thickened to 2–2.5 m, pinching out Layers E and F (Figure 4c), the aggradation of permafrost occurred by the refreezing of a talik formed by the relic thermokarst of a marine terrace.

The thickness of layer A within the built-up area changed from 1.0 to 3.5 m, with an average of 2.2 m. The bottom of thawed layer was crucial to recognizing the distribution and spatial structure of taliks. Concentric taliks with a thickness exceeding 1.8 m (Figure 5a) originated from the infiltration of atmospheric precipitation. Linear taliks either occurred along streets or tributaries of gullies and were formed by thermoerosion. Areas with a thawing depth close to that of natural, undisturbed landscapes (0.5–1.0 m) were either found on the margins of the built-up area under patches of vegetation cover or along the roads paved with ground filling.

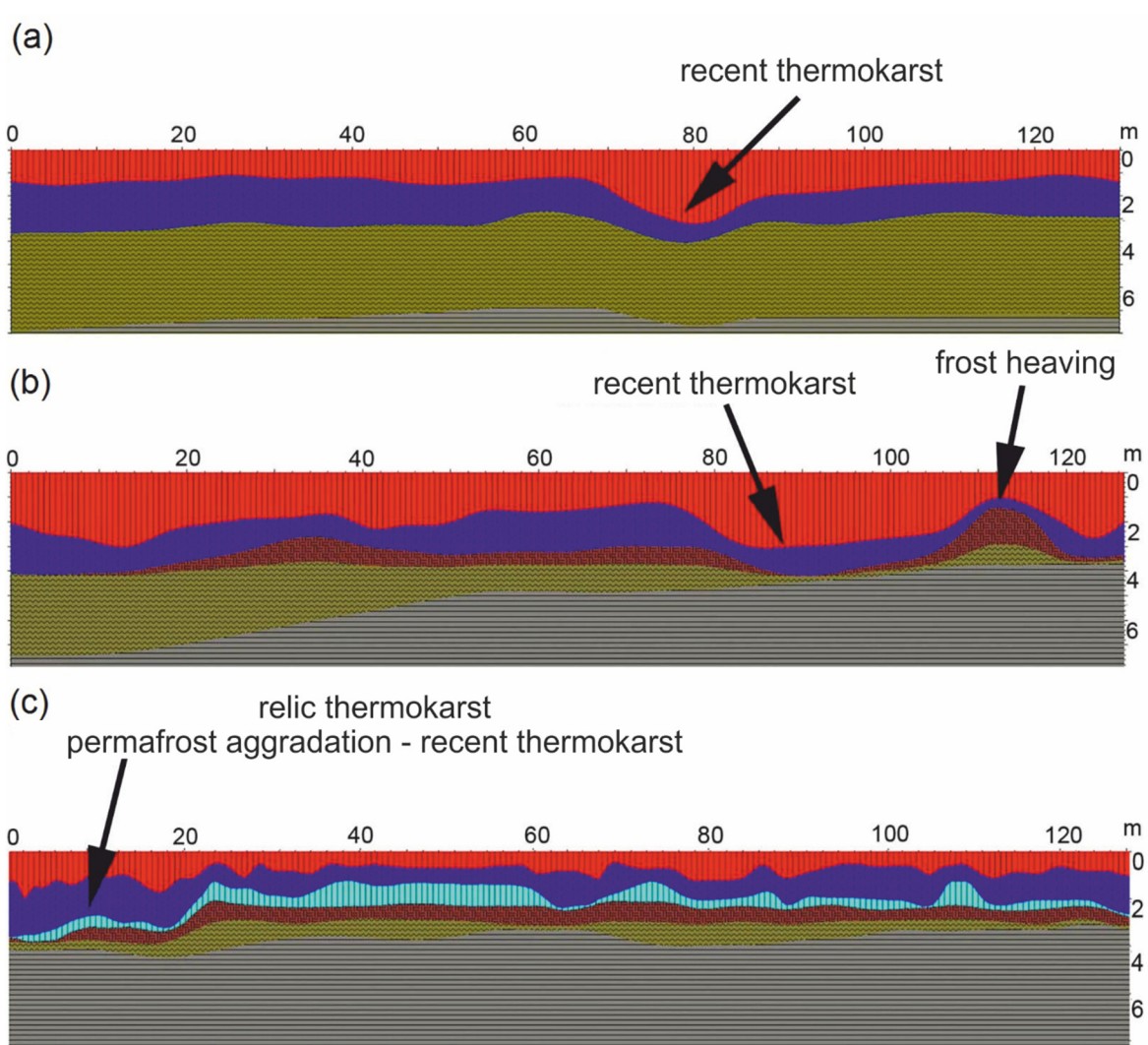

**Figure 4.** Typical traits of cryogenic processes in the sections: thermokarst of the transient layer (**a**), (**b**) and layer E (**c**); frost heaving (**b**); permafrost aggradation (**c**). Refer to the key of Figure 3.

The thickness of the ice-rich Layer B ranged from 0.4 to 3.0 m, with a mean of 1.2 m. The bottom was found at 1.5 to 5.0 m, with a mean of 3.0 m. Covariation of anomalies of the thickness with anomalies of the bottom (Figure 5b) pointed to the aggradation of permafrost by the freezing of relic taliks. The positive anomalies were found in areas with an increased thickness of ground filling: wastelands on sites of demolished constructions, under modern buildings, and in some locations on solifluction slopes on the margins of the village. The thinning of Layer B took place in areas affected by recent ongoing thermokarst or thermal erosion, and also on gradually inundating disturbed lands surrounding the built-up area.

Massive ice beds and organic-rich silts (Layers E and F) were not ubiquitous (Figure 5c,d). Ice beds 0.2–1.5 m thick occurred in the range of depths from 1.0–3.0 m (table) to 2.0–4.0 m (bottom), mostly in the eastern part of the village (Figure 5c). Ice lenses were found in tributaries of gullies surrounding the built-up areas. The bottom negatively correlated with the thickness of Layer E, i.e., thin ice beds were found at 2.5–3.0 m, while thick bodies exceeding 1.0 m were found at 1.5–2.0 m. At the same time, the thickness of Layer E grew with the thickness of underlying Layer F.

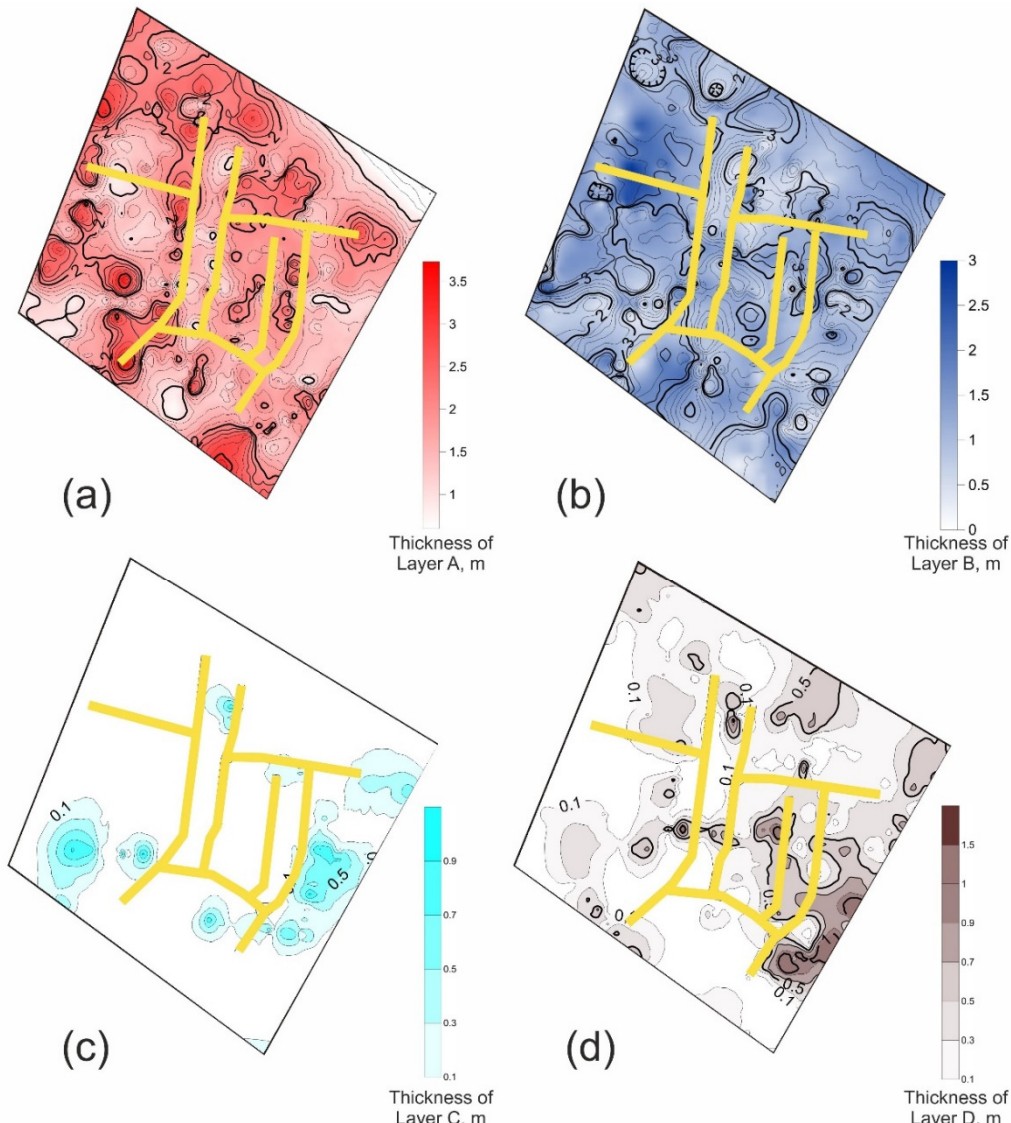

**Figure 5.** Spatial patterns of near-surface layers: (**a**) thickness of Layer A; (**b**) the bottom (isochores) and thickness (filling) of Layer B; (**c**) thickness of Layer E; (**d**) thickness of Layer F. Yellow lines show the network of streets.

Organic-rich, silty, and prone to heaving Layer F formed a continuous cover 0.6–0.8 m thick in central and south-eastern parts of Lorino. Variations of the thickness and the table within the built-up area have formed chains of initiating heaving mounds with ice 1.2–1.5 m thick and with a diameter of 2–3 m, as well as larger heaving areas with ice lenses 8–12 m in diameter (Figure 5d). The bottom of Layer F laid uniformly at 2.7–3.0 m, supporting the frost heaving origin of the anomalies of thickness, rather than filling of the hollows in the underlying sediments.

Based on the interpretation of layers and reconstruction of cryogenic processes, the built-up area of Lorino was classified by the hazard of cryogenic processes into four types (Figure 6). The highest hazard is expected in residential areas, where recent thermokarst or frost heaving were induced by changes of the active layer, especially where Layers E and F lay close to the surface (Figure 4b). It was located along the boundary of ground filling with less disturbed lands surrounding the built-up area. Both processes could be expected to intensify following the tendencies of increasing variability and temperature.

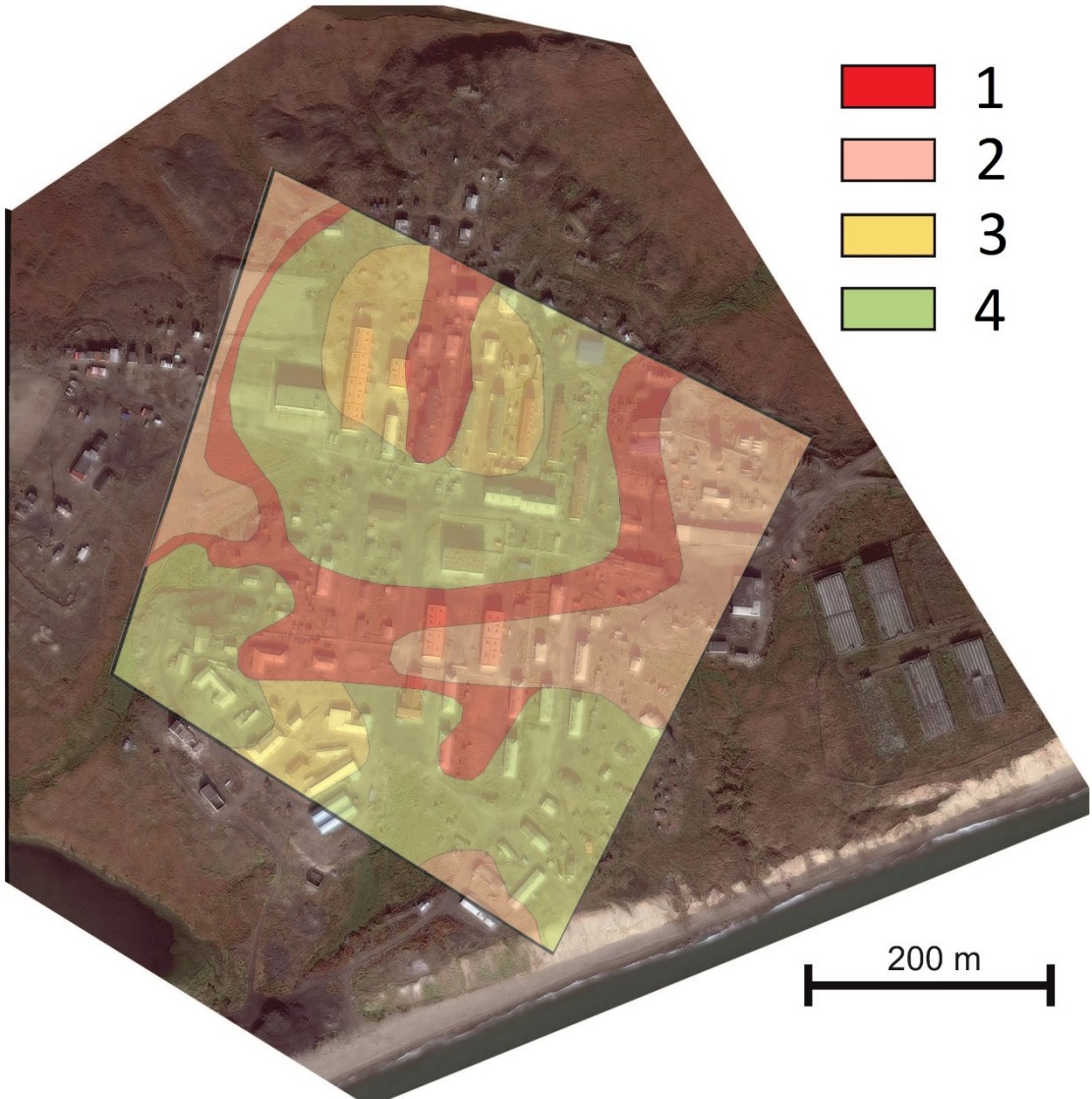

**Figure 6.** Classification of cryogenic hazards in the built-up area of Lorino. Key: (1) areas of active thermokarst and thermoerosion; (2) unstable ice-rich permafrost and heaving-prone sediments; (3) stable permafrost and hazard of secondary thermokarst of Layer B; (4) stable coarse-grained and drained thawed sediments and permafrost.

Overall, at the time of study in 2013, the aggradation of permafrost and formation of the ice-rich layer were observed in the majority of taliks. The intensity of permafrost degradation from thermokarst and erosion was diminished by the new rural design of the 2000s, reconstruction of utilities, and demolition or reconstruction of old buildings. Ground filling supported the rise of the permafrost table and deepened frost susceptible sediments and ice-rich layers, thus decreasing the hazards of frost heaving and thermokarst.

Apart from its practical value, the scheme (Figure 6) illustrates stages of permafrost transition from a natural landscape to a social-ecological system. Permafrost was degraded by the thawing of near-surface ice lenses, but at the same time, it aggraded with an accumulation of the layer of ground filling.

## 4. Conclusions

Our case shows the application of GPR to an in-depth study of geological (cryogenic) processes, phenomena, and hazards in social-ecological systems. To interpret GPR transects as geophysical data,

the descriptions of geological sections are required. The study of etalons prior to spatial surveying helps focus on the variability of a section, rather than on documenting the electrophysical properties. The structure of geophysical layers allows the history of cryogenic processes to be reconstructed. Reconstructing the stages of aggradation and degradation of permafrost and activation of frost heaving extends the application of GPR surveys in cryology and climatology as an effective instrument for palaeoreconstructions.

Current tendencies in the Arctic are likely to cause a growth of hazards of activation of cryogenic processes. In turn, this could increase the deformation of buildings and infrastructure and decrease the quality of life, unless specific measures to control them are undertaken. The application of a GPR survey is a fast and effective method for creating the baseline for planning, and monitoring the efficiency of the measures of adaptation of the Arctic communities to climate change.

An example of an adaptation measure which could be listed among the best practices, when the ice-rich sediments lay close to the surface, is the application of ground filling in Lorino. It is also necessary to keep in mind that the amelioration of permafrost starts with the control of human impacts. Improved maintenance of the buildings and infrastructure, the demolition or reconstruction of old buildings, and repair works of the communication infrastructure in Lorino in the 2000s have created an environment favorable for permafrost aggradation.

Positive lessons from this study might not be applicable at another location due to differences in the geological structure, human impact, and tendencies in climate and environmental change. This is why more case studies addressing the planning and monitoring of adaptation measures are necessary. Additionally, this is why a detailed program of the study was included in this paper. The authors hope it is worth using for improving other permafrost locations.

**Author Contributions:** O.T. developed the methodology; conducted the investigation; completed software processing; and curated, validated, and analyzed data. G.K. secured funding, supervised the study, and administered the project. O.T. and G.K. conceptualized the study, allocated resources, and prepared the original draft. All co-authors prepared the visualization, reviewed and edited the article. All authors have read and agreed to the published version of the manuscript.

**Funding:** This research was funded by the RUSSIAN FOUNDATION FOR BASIC RESEARCH (grant number 18-05-60036), UK FCO PROSPERITY FUND (PPY RUS 1308-1401), and the STATE ASSIGNMENTs to the Institute of Physico-Chemical and Biological Problems in Soil Science of the Russian Academy of Sciences (grant number 0191-2019-0044) and to the Laboratory of Geoecology of Northern Territories (project AAAA-A16-116032810055-0). The APC was funded by the Russian Foundation for Basic Research, G. Kraev and A. Maslakov.

**Acknowledgments:** The authors thank the administration of Chukotka District of Chukotka Autonomous Okrug, Russia, and personally thank M.A. Zelenskii and N.L. Kalyanto for the provision of access to archive materials of geological surveys. We also thank the mayor of Lorino V.N. Kalashnikov for logistical support. The authors also thank two anonymous reviewers for valuable comments, which helped significantly improve the quality of our article.

**Conflicts of Interest:** The authors declare no conflicts of interest. The funders had no role in the design of the study; in the collection, analyses, or interpretation of data; in the writing of the manuscript; or in the decision to publish the results.

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
