# Peer review of "Hazards of Activation of Cryogenic Processes in the Arctic Community: A Geopenetrating Radar Study in Lorino, Chukotka, Russia"

_geosciences, doi:10.3390/geosciences10020057_

Round 1
Reviewer 1 Report
The topic of this paper is interesting and important in the context of increased vulnerability of the cryosphere due to climate changes. The paper indicates for the first time the structure of permafrost and the cryogenic processes in the Chukotka Russia, by means of geophysical measurements. This study gives interesting insights into the local characteristics of active layer thickness and the precise location of thermokarst features or frost heaving.
In general GPR is not the best tool in detecting permafrost conditions; however I agree, it is used in permafrost environment frequently. Although the GPR technique often reveals less information regarding the substrate in permafrost terrain compared with ERT or seismic, in some situations it provides valuable complementary data.
Because of the overall quality of the work, this paper is suitable for a publication in Geosciences however, to my opinion, several important points need to be developed, corrected and/or modified before acceptation:
The paper is clearly structured and the English is fairly good, but needs improvement by a native speaking person. Within the Introduction the aim and objectives of the paper are not clearly stated. The presentation of the study area and the state of the art should be improved. There are no/vague information on the climatic conditions, geomorphology of the area and permafrost conditions at this site. There is no indication of previous similar studies in the region. Please clarify this aspect. The methodological part should include more details on GPR data processing. Some of the literature cited here is quite outdated and should be replaced with more recent/relevant publications (see suggestions below) on GPR measurements in permafrost areas. Permafrost is a thermal state of the ground, soil, rock etc. Using GPR the radargrams might only reveal the presence of ground ice and not the thermal conditions of the soil and therefore the active layer characteristics. The soil might have a negative temperature, without containing ground ice and in this case, the GPR will not be able to reveal the depth of the active layer. Please refer to this in the text. Please briefly refer to the reflections you identified in the radargrams. Are they parallel? Did you identified hyperbolas or stacked reflectors? In which layer the pattern of reflections is dense and in which is a lack of reflections?References:
Petrone, J., Sohlenius, G., Johansson, E., Lindborg, T., Näslund, J-O., Strömgren, M., Brydsten, L., 2016. Using ground-penetrating radar, topography and classification of vegetation to model the sediment and active layer thickness in a periglacial lake catchement, western Greenland. Eart Syst. Sci. Data, 8, 663-667.
Ardelean, A.C., Onaca, A., Urdea, P., Sărășan, A., 2017. Quantifying postglacial sediment storage and denudation rates in a small alpine catchment of the Făgăraș Mountains (Romania). Science of the Total Environment, 599-600, 1756-1767.
Campbell, S., Affleck, R.T., Sinclair, S., 2018. Ground-penetrating radar studies of permafrost, periglacial and near-surface geology at McMurdo Station, Antarctica. Cold Regions Science and Technology, 148, 2018, 38-49.
Onaca, A., Ardelean, F., Ardelean, A., Magori, B., Sirbu, F., Voiculescu, M., Gachev, E., 2020. Assessment of permafrost conditions in the highest mountains of the Balkan Peninsula. Catena, 185, 104288.
Specific comments are given in the attached document.

Author Response
We thank the reviewer for high appreciation of our study. Reviewer's comments and focus on the shortcomings of the manuscript helped significantly improve our paper. All newly introduced text is marked yellow in the attached manuscript review file.
We would like to stress thath although GPR is not the best in permafrost conditions elsewhere compared to other geophysical methods, there are several advantages in using it in the built-up area, which are listed in the Section 3.3.
We have reviewed the paper, improved the text and passed the English correction serviceses provided by MDPI.
We have added more information on morphology, climate and permafrost conditions to the study site section (Section 2.1).
We have vlearly stated that to our knowledge there are no previous geophysical data for the study site, and made references to other GPR studies in Chukotka, made by dr. O.D. Tregubov.
We did not replace the ''old'' literature, because we use the experience of our predecessors using GPR for permafrost studies. However we included references to the several new studies suggested by reviewer.
Following the reviewers comments we significantly reworked the Methods Section 2.2: general principles of operation of GPR and processing of radargrams were included, and detailed information on four stages of the study, including Reconaissance; Etalon compilation; Collection and processing of transects; Spatial modeling
We would like to assure (and that is visible on our Figure 3) reviewer, that survey was produced by an experienced operator, and parallel reflections were filtered during processing. Hyperbolas observed in subsurface originated from buried construction wastes and communications and were neglected in our study according to our goals and objectives.
We again thank reviewer for his time and work.
If MDPI allows and if the Reviewer agrees, we would be glad to acknowledge him/her personally.

Reviewer 2 Report
Dear Editors,
It is interesting and useful investigation.
However, I have several minor remarks:
(1) Figure 4. Not "frost heving", but "frost heaving".
(2) English must be slightly smoothed.
(3) Authors of he paper do not refer to key papers of the thermal peculiarities of permafrost:
Kutasov, I.M. and Eppelbaum, L.V., 2017. Time of Refreezing of Surrounding the Wellbore Thawed Formations. International Journal of Thermal Sciences, 122, 133-140
Kutasov, I.M. and Eppelbaum, L.V., 2018. The effect of thermal properties changing (at ice-water transition) on the radius of permafrost thawing. Cold Regions Science and Technology, 151, 156-158.
Eppelbaum, L.V. and Kutasov, I.M., 2019. Well drilling in permafrost regions – dynamics of the thawed zone. Polar Research, 38, No. 2, 1-9.
After the aforementioned small revision, this MS can be accepted for publication.
Author Response
We would like to thank the reviewer for his positive appraisal of our study, the time it took to review it and the valuable comments, which helped to improve our article.
We have corrected the text and figures. Rewrote parts of the text and introduced the new ones following the comments of both reviewers. Also the langauage improvement service of MDPI have corrected the grammar.
Because our paper is focused on geophysics we did not find it applicable to use the references suggested by the reviewer in our study. However for the remarks on disturbing means of drilling of permafrost we used another link with one of the co-authors of the proposed studies to reference:
Mel'nikov, P.I.; Balobayev, V.T.; Kutasov, I.M.; Devyatkin, V.N. Geothermal studies in Central Yakutia. International Geology Review 1974, 16, 565-568, doi:10.1080/00206817409471838.
